# NN-Baker: A Neural-network Infused Algorithmic Framework for Optimization Problems on Geometric Intersection Graphs

**Evan McCarty** *
Department of Computer Science
University of Illinois, Chicago
emccarty@uic.edu

**Qi Zhao** *
Computer Science and Engineering Department
University of California, San Diego
qiz032@ucsd.edu

**Anastasios Sidiropoulos**
Department of Computer Science
University of Illinois, Chicago
sidiropo@uic.edu

**Yusu Wang**
Halıcıoğlu Data Science Institute
University of California, San Diego
yusuwang@ucsd.edu

## Abstract

Recent years have witnessed a surge of approaches to use neural networks to help tackle combinatorial optimization problems, including graph optimization problems. However, theoretical understanding of such approaches remains limited. In this paper, we consider the geometric setting, where graphs are induced by points in a fixed dimensional Euclidean space. It turns out that several graph optimization problems can be approximated (in a bicriteria manner) by an algorithm that runs in time linear in graph size $n$ via a framework that we call the Baker-paradigm. A key advantage of the Baker-paradigm is that it decomposes the input problem into (at most linear number of) small sub-problems of **bounded sizes** (independent of the size of the input). For the family of such bounded-size sub-problems, we can now design neural networks with universal approximation guarantees to solve them. This leads to a mixed algorithmic-ML framework, which we call NN-Baker that has the capacity to approximately solve a family of graph optimization problems (e.g, maximum independent set and minimum vertex cover) in time linear in the input graph size. We instantiate our NN-Baker by a CNN version and GNN version, and demonstrate the effectiveness and efficiency of our approach via a range of experiments.

## 1 Introduction

Many tasks in science and engineering can be naturally modeled by combinatorial optimization problems over graphs, such as maximum independent set, minimum vertex cover, minimum multi-way cut, maximum clique, and so on. These problems are often NP-hard. Hence there has been great effort devoted to developing efficient approximation algorithms. However, many such problems are hard to approximate in the general setting as well: for example, it is known that the maximum independent set problem is NP-hard, and it is even NP-hard to approximate within $n^{1-\varepsilon}$ on $n$-vertex graphs, for any fixed $\varepsilon > 0$ [1].

On the other hand, many of these hard problems admit PTAS in the *geometric setting* when the graphs are assumed to be induced by points in fixed-dimensional Euclidean space. Here, a *PTAS*

---

*Equal contribution

35th Conference on Neural Information Processing Systems (NeurIPS 2021).

*(polynomial-time approximation scheme)* is a polynomial time algorithm which for any fixed $\varepsilon > 0$, can approximate a maximization problem within a factor of $(1 + \varepsilon)$ (or within a factor of $(1 - \varepsilon)$ for a minimization problem). One most prominent example is the travelling salesman problem (TSP), where in general there is no PTAS available (and currently the best known approximation algorithm achieves a factor of $3/2 - \alpha$, for some $\alpha > 10^{-36}$ [2]). However a PTAS, more specifically in this case a near linear time $(1 + \varepsilon)$-approximation algorithm, was developed for the Euclidean TSP problem in a ground breaking work by Arora [3]. The maximum independent set problem is known to remain NP-hard in the geometric setting [4], but it is known that it also admits a PTAS [5, 6, 7].

Nevertheless, for problems of large size, even these PTAS are still not practical, especially since many of them involve large constants that are **exponential** in some fixed parameter (e.g, having a term like $n^{\frac{1}{\varepsilon}}$ where $(1 + \varepsilon)$ is the approximation factor). In applications, practitioners often rely on handcrafted heuristic search strategies to find high-quality solutions.

Recently, there has been a surge on using machine learning (ML) for combinatorial optimization problems; see the review paper by Bengio *et al.* [8] which also provides a nice categorization of different ways that a ML component may contribute to the combinatorial optimization problems. Earlier such approaches focus on several graph combinatorial optimization problems, including TSP, minimum vertex cover and maximum independent sets; see e.g., [9, 10, 11, 12, 13, 14, 15, 16]. More recently, there has been a range of approaches developed to tackle more general mixed integer linear programming problems (MILP); e.g, [17, 18, 19, 20, 21]. See Section 1.1 for more detailed description of some related work.

In general, an ML framework can be trained to solve an optimization problem in *an end-to-end manner* (e.g, [10, 11, 12, 15, 16]). Alternatively, many recent approaches use ML module as a component *within a specific algorithmic framework* to help make hopefully better (either in terms of quality or efficiency) decisions during the execution of this algorithm: such as using imitation learning or reinforcement learning to learn a good policy to decide which variable to branch in a branch-and-bound algorithm [18, 20]. Despite the tremendous progress in combining ML with optimization problems, theoretical understanding remains limited: Does a proposed ML pipeline have the expressiveness capacity to solve the combinatorial optimization problem exactly or approximately? What is a suitable model for input data distribution to talk about generalization?

Consider the capacity question: While neural networks are known to have many universal approximation results for real or complex valued continuous functions (or some other special families of functions), e.g, [22, 23, 24, 25], combinatorial optimization problems, say finding the maximum independent set, cannot be easily modeled to fit into such function approximation frameworks. Nevertheless, a very interesting recent work [13] shows that for graph combinatorial optimization problems, the so-called vector-vector consistent graph neural networks ($VV_C$-GNNs), can solve the same family of problems as the distributed local algorithms in the so-called port-numbering models. Leveraging the literature in distributed local algorithms [26, 27], this leads to several positive results on the capacity of GNNs for approximating minimum vertex cover or maximum matching problems with certain constant factors, however **only for graphs with bounded degrees** – intuitively, the depth of the GNN will depend on the bound $\Delta$ on the maximum node degree of input graphs. Unfortunately, the connection to distributed local algorithms also leads to negative results: roughly speaking, these constant factor approximations for the special family of **constant-degree graphs** are the best that a GNN can do for these graph optimization problems. See [13] for details.

**Our work.** The aforementioned results (on the capacity of GNNs) are for the case where GNNs are used to solve an optimization problem *in an end-to-end manner*, and other than the bound on max-degree, the input graphs are abstract graphs. In this paper, we advocate the study of using ML for optimization problems in the *geometric setting* where (graph) optimization problems are induced by points in a fixed dimensional Euclidean space $\mathbb{R}^d$. This is a rather common scenario in practice, such as solving TSP in road networks spanned by cities in $\mathbb{R}^2$, or solving maximum independent set in a communication network spanned by sensor nodes in $\mathbb{R}^2/\mathbb{R}^3$ (i.e, the unit-ball model that we will introduce in Section 2.1). Such graphs can also be the result of an embedding of an input arbitrary graph into a certain latent space. At the same time, the geometric setting brings special structures to the problem at hand, which an algorithm and also ML can then leverage.

In particular, using the maximum independent set (MIS) problem as an example, we first propose what we call *the Baker-paradigm* in Section 2, which is an approximation framework for geometric

optimization problems inspired by Baker's work in [28]. We show that the Baker-paradigm gives a bi-criteria approximation for the MIS in the Euclidean setting (Theorem 2.1). The running time is only *linear* in the size of input point set, but *exponential* in terms of the parameters. This framework is general and can be extended to several other geometric optimization problems (Section 2.3).

A key advantage of our Baker-paradigm is that it decomposes the problem into (at most a linear number of) small sub-problems of fixed size (independent of size of input graphs). For the family of such fixed-size sub-problems, we can now design neural networks with universal approximation guarantees to solve them. Using such a neural network to replace (Step-2) of our Baker-paradigm, we then obtain a mixed algorithmic-ML framework, which we refer to as NN-Baker, that has the capacity to produce a bi-criteria approximation of MIS within *any* constant factor in near linear time (i.e, a bi-criteria PTAS); see Section 3.1 and Theorem 3.2. Note that while Theorem 2.1 already gives a near-linear time bi-criteria PTAS for MIS, the constant involved in the time complexity is exponential in the approximation parameters, making it inefficient in practice. In contrast, the NN-Baker will replace the costly component by a neural network component, and only call this neural network at most $n$ times. Other than calls to neural networks, the time needed is $\Theta(n)$ where the constant contains only terms polynomial in the approximation paramters. The resulting mixed algorithmic-ML framework is very efficient, as we show in Section 4.

In Section 3.2, we provide two instantiations of the NN component for our NN-Baker: based on CNN and GNN respectively. We present a range of experimental results in Section 4 to show the effectiveness of both the CNN-Baker and GNN-Baker frameworks. Note that our NN-Baker can be used together with other SOA NN framework to solve combinatorial optimization problems and to further improve them (sometimes significantly). For example, we deploy different SOA GNNs for graph optimization problems to instantiate (Step-2) in NN-Baker, including TGS of [12] in the supervised setting, LwD of [16] in the reinforcement learning setting, as well as Erdős-GNN [15] in the unsupervised setting. We show that as the problem size increases, the performance of each original GNN decreases (the GNN is always trained and tested on problems of similar sizes). However, using GNN-Baker significantly improves the performance of the original GNN as the problem size increases – This is partly because, independent of the problem size, in (Step 2) our GNN-Baker only needs to train and test the GNN component on a small graph (of bounded size). Thus the trained GNN can adapt to the problem structure much better and require much fewer training samples.

Our NN-Baker is, to our best knowledge, the first (bi-criteria) PTAS for a combinatorial problem for an ML-based approach (in terms of expressiveness). The recent line of work of using a ML component (e.g a GNN trained by imitation learning) to make branching decisions within the branch-and-bound algorithmic framework [18, 20] may solve the exact problem *given enough running time*. However, the number of times the algorithm calls the NN component may be exponential in the input size. Instead, our NN-Baker framework calls the neural network (which has a bounded size) only a linear number of times. Our approach can open new directions to design NN-infused algorithmic frameworks with theoretical guarantees, by for example, leveraging divide and conquer paradigm and replacing certain algorithmic components by neural components.

## 1.1 More on related work

The idea of using neural networks to tackle optimization problems traces back to the 1980's. One of the most important frameworks in this direction is the Hopfield Neural Network (HNN) [29, 30]. In particular, the HNN is a feedback neural network with pre-specified weights (whose assignments depend on the optimization problem at hand), which encodes a dynamic system whose associated energy function characterize the optimization problem to be solved. To use it to solve an optimization problem, one starts with an initial state, and iterates till convergence.

As branch-and-bound (B&B) has been proven to be a powerful framework in solving optimization problems, especially in MILP (mixed integer-linear programing) problems, researchers proposed different machine learning algorithms to boost B&B. [17] provided a list of features of variables and designed a learning-to-rank model on selecting branching variables. [18, 20, 21] developed GNN approaches to learn the policy of choosing branching variables after formulating MILP problems as graphs by imitation learning or reinforcement learning. Besides MILP problems, GNNs are also applied on graph combinatorial optimization problems. [9] encodes the input graphs as sequences and takes an attention RNN to process the sequences. It can be used to compute convex hulls or solve problems like TSP. [10, 11, 31, 32, 16] takes reinforcement learning on graphs to solve routing

problems like TSP, and other problems like maximum independent set. [12] solves graph theory problems by supervised learning setup after solving a set of cases as training set by existing solvers, while [15] introduces unsupervised approaches by designing loss function and training setup based on objective functions and variables constraints. In addition, there are works [13, 33, 34] that study the power of GNN on solving different kinds of combinatorial optimization problems.

## 2 The Baker-paradigm

We now propose a framework to obtain approximation algorithms for certain geometric optimization problems based on the work of Baker [28]. Baker's technique has been applied successfully to a plethora of optimization problems on planar graphs, including Maximum Independent Set, Minimum Vertex Cover, Minimum (Edge) Dominating Set, Maximum Triangle Matching, Maximum H-Matching, TSP, and many others. Furthermore, the technique has been adapted to the geometric setting (see [35] for a survey). From the geometric setting, the most relevant work to ours is [36], where the authors obtain approximation algorithms for maximum independent set and minimum vertex cover on unit disk graphs. We will show in the next section how neural networks can be used to obtain efficient algorithms within this framework. We begin with some definitions.

### 2.1 Preliminaries

We consider the geometric setting: i.e., combinatorial optimization problems on geometric intersection graphs. We present our methods for the case when the input is a **unit-ball graph**, which is the intersection graph of unit balls in $\mathbb{R}^d$. Specifically, graph nodes correspond to a set of balls of unit radius in $\mathbb{R}^d$, and two nodes are connected by an edge iff the two balls intersect. Our approach can be extended to intersection graphs of several other geometric objects such as unit hypercubes, ellipsoids of bounded aspect ratio, and so on. For the sake of succinctness, we focus on the case of unit balls.

**Approximations and bi-criteria relaxations.** The algorithms we will develop are bi-criteria approximations. The precise definition of a bi-criteria optimization problem depends on the space of feasible solutions. For concreteness, let us focus on the *d-dimensional Maximum Independent Set of Unit Balls problem* (denoted by $d$-**MIS**), for some fixed dimension $d \in \mathbb{N}$: The input to the $d$-MIS problem is a set of points $X \subset \mathbb{R}^d$, which corresponds to the set of centers of unit balls. Let $G_X = (X, E)$ denote the intersection graph spanned by points in $X$, where $(x, x') \in E$ if the unit balls ball$(x, 1)$ and ball$(x', 1)$ intersect (meaning that $\|x - x'\| \le 2$). The goal is to find the maximum independent set of $G_X$, which is equivalent to finding some maximum cardinality subset of disjoint balls centered at $X$. Let $\mathsf{OPT}(X)$ denote the size of a maximum independent set for $G_X$. For any $\alpha > 0$, an algorithm is an $\alpha$-*approximation for* $d$-MIS if on any input $X \subset \mathbb{R}^d$, it outputs some independent set $Y \subseteq X$, with $\mathsf{OPT}(X)/\alpha \le |Y| \le \mathsf{OPT}(X)$.

The bi-criteria version of the problem is defined as follows. Let $\varepsilon > 0$. We say that some $Y \subseteq X$ is $(1 + \varepsilon)$-*independent* if the balls of radius $1/(1 + \varepsilon)$ centered at the points in $Y$ are disjoint; that is, for all $p, q \in Y$, we have $\|p - q\|_2 > 2/(1 + \varepsilon)$. We denote the size of the maximum $(1 + \varepsilon)$-independent subset of $X$ by $\mathsf{OPT}_{1+\varepsilon}(X)$. For any $\alpha \ge 1$, $\beta \ge 1$, We say that an algorithm is $(\alpha, \beta)$-*bi-criteria approximation* if on any input $X \subset \mathbb{R}^d$, outputs some $\beta$-independent set $Y \subseteq X$, with $\mathsf{OPT}(X)/\alpha \le |Y| \le \mathsf{OPT}_\beta(X)$.

**Randomization.** The algorithms we present for $d$-MIS are randomized, and thus the size of the output is a random variable. We use the following standard extensions of the above definitions in this setting. We say that a randomized algorithm is $\alpha$-approximation in expectation if on any input $X$ it outputs a solution $Y$ with $\mathsf{OPT}(X)/\alpha \le \mathbf{E}[|Y|] \le \mathsf{OPT}(X)$. We say that a randomized algorithm is $(\alpha, \beta)$-bi-criteria approximation in expectation if on any input $X$ it outputs a solution $Y$ with $\mathsf{OPT}(X)/\alpha \le \mathbf{E}[|Y|] \le \mathsf{OPT}_\beta(X)$.

### 2.2 Baker's paradigm for $d$-MIS

In this section, we describe our Baker-paradigm to obtain an approximation algorithm for $d$-MIS. We will later see that the same method can be extended to several other optimization problems on geometric intersection graphs. Let $\varepsilon > 0$ be arbitrarily small but fixed. The algorithm proceeds in the following steps: See Figure 1 for an illustration.

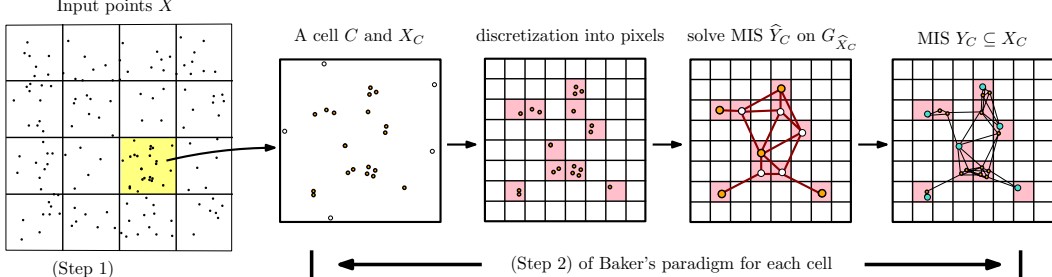

Figure 1: Illustration of Baker-paradigm in 2D. First we put a randomly shifted grid on input points $X$. Consider a cell $\mathsf{C}$, empty dots close to boundary are $X'_\mathsf{C}$. We then snap points in $X_\mathsf{C} \setminus X'_\mathsf{C}$ into pixels in a second level grid; and those dark (non-empty) pixels form $\widehat{X}_\mathsf{C}$. The MIS $\widehat{Y}_\mathsf{C}$ (bigger orange nodes) is then computed on the graph $G_{\widehat{X}_\mathsf{C}}$ spanned by dark pixels $\widehat{X}_\mathsf{C}$; note that for clarity of illustration, in the 4-th picture we use the center of each pixel to represent this pixel (instead of its bottom-left corner). The solution $\widehat{Y}_\mathsf{C}$ is then converted to a MIS $Y_\mathsf{C} \subseteq X_\mathsf{C} \setminus X'_\mathsf{C}$. In (Step 3), the union of all $Y_\mathsf{C}$ for all cells is returned as the final MIS.

**Algorithm Baker-**MIS: The input is a set of points $X \subset \mathbb{R}^d$, with $|X| = n$, where each point in $X$ is the center of a unit ball. The algorithm has three steps.

**Step 1: The randomly shifted grid.** Let $\Gamma$ be an axis-parallel hyper-grid, where each cell is a $d$-D axis-parallel hypercube of side-length $\mathsf{r} = \frac{2d}{\varepsilon}$. Specifically, let $\Gamma = \bigcup_{i=1}^{d} \bigcup_{j \in \mathbb{Z}} \{h_i + j\frac{2d}{\varepsilon}e_i\}$, where $e_1, \ldots, e_d$ is the standard orthonormal basis, and for $i \in [1, d]$, $h_i$ is the $(d\text{-}1)$-D hyperplane that passes through the origin and is orthogonal to $e_i$. Pick $\tau \in [0, 2d/\varepsilon)^d$ uniformly at random. Let $\Gamma + \tau$ denote the grid obtained by shifting $\Gamma$ by the vector $\tau$.

**Step 2: Bi-criteria solution for the problem locally on each cell.** Now given a cell $\mathsf{C}$ of $\Gamma + \tau$, for any $P \subset \mathbb{R}^d$, let $P_\mathsf{C} = P \cap \mathsf{C}$ denote the restriction of $P$ within $\mathsf{C}$. Let $X'$ be the set of centers of the unit balls in $X$ that intersect the shifted grid $\Gamma + \tau$ (i.e., these are points within distance-1 to the hyperplanes (or gridlines for the 2D case) in the grid). Let $\delta > 0$, and set $\delta' = \delta/\sqrt{d}$. We partition **each cell** $\mathsf{C}$ of $\Gamma + \tau$ to a $d$-dimensional grid of pixels, where each pixel is a $d$-dimensional hypercube of side length $\delta'$. We snap each point in $X \setminus X'$ to the corner of the pixel containing it that is closest to the origin, thus obtaining the set $\widehat{X}$, ignoring multiplicities; that is

$$\widehat{X} = \bigcup_{(p_1, \ldots, p_d) \in X \setminus X'} \{(\delta'\lfloor\tfrac{p_1}{\delta'}\rfloor, \ldots, \delta'\lfloor\tfrac{p_d}{\delta'}\rfloor)\}.$$

Now for each cell $\mathsf{C}$ of $\Gamma + \tau$, $\widehat{X}_\mathsf{C}$ (the restriction of $\widehat{X}$ to $\mathsf{C}$) consists of the bottom-left corners of those pixels containing some points in $X_\mathsf{C} \setminus X'_\mathsf{C}$. Let $G_{\widehat{X}_\mathsf{C}}$ be the intersection graph of the radius-$(1 - \delta)$ balls with centers in $\widehat{X}_\mathsf{C}$. Note that the size of $\widehat{X}_\mathsf{C}$ is at most $(\frac{2d}{\varepsilon\delta'})^d$, and thus the graph $G_{\widehat{X}_\mathsf{C}}$ is of bounded size. Furthermore, note that any independent set of $G_{\widehat{X}_\mathsf{C}}$ can be of size at most $s = V_d(\frac{2d}{\varepsilon})^d$ (due to a simple packing argument), where $V_d$ denotes the volume of the $d$-dimensional ball of radius $(1 - \delta)$. We can then compute the maximum independent set $\widehat{Y}_\mathsf{C} \subseteq \widehat{X}_\mathsf{C}$ in $G_{\widehat{X}_\mathsf{C}}$ by a brute-force enumeration of all subsets of $\widehat{X}_\mathsf{C}$ of cardinality at most $s$, and returning the maximum cardinality subset that is independent in $G_{\widehat{X}_\mathsf{C}}$. Finally, we compute $Y_\mathsf{C} \subseteq X_\mathsf{C}$ by mapping each point $\widehat{p} \in \widehat{Y}_\mathsf{C}$ to an arbitrary point $p \in X_\mathsf{C} \setminus X'_\mathsf{C}$ that lies in the pixel that $\widehat{p}$ represents.

**Step 3: The final solution** is $Y = \bigcup_\mathsf{C} Y_\mathsf{C}$, the union of MIS returned within all non-empty cells.

The proof of the following theorem can be found in the Supplement.

**Theorem 2.1.** *Let* $\varepsilon, \delta > 0$. *The algorithm Baker-*MIS *is a* $(1 + \Theta(\varepsilon), 1 + \Theta(\delta))$-*bi-criteria approximation in expectation for* MIS. *On input a set of size* $n$, *the algorithm runs in time* $(1/(\varepsilon\delta))^{(d/\varepsilon)^{O(d)}} n$.

Recall that our algorithm removes those points $X'$ within distance 1 to the grid. Intuitively, we need randomly shifted grid so that this $X'$ does not contain too many points from an optimal maximum

independent set in expectation. Note that in practice, we can further improve the quality of the output: Specifically, currently, all points within distance 1 to the shifted grid $\Gamma + \tau$ (i.e, $X' \subseteq X$) are removed. The reason is to ensure that solutions (max-independent sets) of neighboring cells do not conflict each other. We can add some of those points from $X'$ back to the solution $Y$ as long as they do not cause conflict (i.e, within distance 2) to any points in $Y$. We can do so in a greedy manner in practice to obtain an even better solution. Our theorem above holds for this greedily improved solution as it remains an independent set and is a superset of $Y$.

**Removing the Bi-Criteria Condition.** We remark that directly using the original Baker idea one can in fact obtain a $(1 + \varepsilon)$-approximation for MIS (instead of a bi-criteria approximation), by not having a second-level discretization in each cell in (Step 2). This is effectively a randomized version of the algorithm from [36]. The standard details can be found in the Supplement. As shown in the following theorem, the price to pay is that the dependency of time complexity on $n$ increases from previous $n$ (i.e, linear) to $n^{(1/\varepsilon)^{O(d)}}$, which is significant.

**Theorem 2.2.** *We can modify Baker-paradigm to provide a $(1 + \Theta(\varepsilon))$-approximation in expectation for* MIS. *On input a set of size $n$, this modified algorithm runs in time $n^{(1/\varepsilon)^{O(d)}}$ .*

## 2.3 Other graph optimization problems

We now briefly discuss how the above general framework can be extended to other problems on unit ball graphs, with only minor modifications. We describe some representative such problems.

**Minimum vertex cover.** In the Minimum Vertex Cover (MVC) problem, we are given a graph $G$ and the goal is to find a minimum cardinality set $U \subseteq V(G)$, such that all edges in $G$ have at least one endpoint in $U$. In the case were $G$ is a unit ball graph in $\mathbb{R}^d$, this problem can be solved by modifying the Baker paradigm as follows. In Step 2, we enlarge each cell C of $\Gamma + \tau$ by increasing its side length to $2 + 2d/\varepsilon$. Thus, any two adjacent cells have an intersection of width 2, and some points in $X$ may fall in multiple cells. By the linearity of expectation, it follows that the expected number of points in any optimal solution that fall in multiple cells (counting multiplicities) is at most $\varepsilon|\mathsf{OPT}(X)|$. Therefore, by solving the problem independently on each cell and taking the union of all the solutions we obtain a $(1 + \varepsilon)$-approximate solution for the initial problem $X$. Discretizing each cell into further pixels gives rise to a more efficient, but bi-criteria approximation.

**Maximum Acyclic Subgraph, Planar Subgraph, and $\mathcal{F}$-Minor Free Subgraph.** In the Maximum Acyclic Subgraph problem we are given a graph $G$ and the goal is to compute a subgraph of $G$ with a maximum number of vertices that is acyclic. It follows by the linearity of expectation, that the expected number of balls in any optimal solution that intersect the randomly shifted grid is at most $\varepsilon|\mathsf{OPT}(X)|$. Thus, solving the problem on each cell and taking the union of all the acyclic subgraphs found, results in a $(1 + \varepsilon)$-approximate optimal acyclic subgraph of the input. (Similar to MVC, discretizing each cell into further pixels gives rise to a more efficient, but bi-criteria approximation.) The exact same argument works also for the Maximum Planar Subgraph problem, where the goal is to find a subgraph with a maximum number of vertices that is planar. Finally, the same argument extends to the case of the more general Maximum $\mathcal{F}$-Minor Free Subgraph problem, where the goal is to find a subgraph with a maximum number of vertices that does not contain as a minor any of the graphs in a fixed family $\mathcal{F}$. We note that this problem generalizes the Maximum Acyclic Subgraph problem (when $\mathcal{F}$ contains the triangle graph) and the Maximum Planar Subgraph problem (when $\mathcal{F}$ contains $K_5$ and $K_{3,3}$).

## 3 A NN-Baker framework

### 3.1 Infusing neural network inside the Baker-paradigm

Instead of solving (Step 2) of Baker-paradigm in a brute-force manner, we can replace it by a neural network, and we refer to the resulting generic paradigm as *NN-Baker*. Roughly speaking, we will replace the exact computation of a MIS in Step 2 of algorithm Baker-MIS by a neural network. More specifically, consider the following:

**Step 2'.** We follow the same notations as in Step 2 of algorithm Baker-MIS. For each cell C of the grid $\Gamma + \tau$, we proceed as follows. Recall $\widehat{X}_\mathsf{C}$ is a set of corners of all non-empty pixels

(i.e, containing some point from $X \setminus X'$) in C. Let $W_C$ be the set of all pixels in C, and let $\mathcal{P}_C$ be the powerset of $W_C$. Let $f_{\mathrm{MIS}} : \mathcal{P}_C \to \mathcal{P}_C$ be such that for all $Z \in \mathcal{P}_C$, $f_{\mathrm{MIS}}(Z)$ is some optimal solution to the MIS problem on input $Z$ w.r.t. radius $(1 - \delta)$; that is, $f_{\mathrm{MIS}}$ maps an instance to an optimal MIS solution for the intersection graph formed by radius $(1 - \delta)$ balls. Since every point in $X_C \setminus X'_C$ is at distance at most $\delta$ from some point in $\widehat{X}_C$, it follows that $f_{\mathrm{MIS}}(\widehat{X}_C)$ is a $(1, 1 + \Theta(\delta))$-bi-criteria solution for the set of points in $X_C \setminus X'_C$ (see the argument in the proof of Theorem 2.1 in Supplement). We can view $f_{\mathrm{MIS}}$ as a mapping between indicator vectors of subsets of $W_C$; i.e. $f_{\mathrm{MIS}} : \{0, 1\}^k \to \{0, 1\}^k$, where $k = |\mathcal{P}_C| = 2^{(2d/(\varepsilon\delta))^d}$. Let $\widehat{f}_{\mathrm{MIS}} : [0, 1]^k \to [0, 1]^k$ be any continuous extension of $f_{\mathrm{MIS}}$. We then approximate $\widehat{f}_{\mathrm{MIS}}$ by a function $g_{\mathcal{N}} : [0, 1]^k \to [0, 1]^k$ as computed by a neural network $\mathcal{N}$. We round coordinate-wise the output of $g_{\mathrm{MIS}}$ to a vector in $\{0, 1\}^k$, by setting every value greater than $1/2$ to 1, and all other values to 0. We thus obtain the indicator vector of some $Y_C \subset W_C$. Alternatively, we can produce a discrete solution by the following greedy strategy: We sort the points in $W_C$ in non-increasing order of their values in the vector $g_{\mathcal{N}}(\widehat{X}_C)$, and we take $\widehat{Y}_C$ to be a maximal prefix of this sorted order that forms an $(1 + \Theta(\delta))$-independent set in $G_{\widehat{X}_C}$. Finally, we compute $Y_C \subseteq X_C \setminus X'_C$ by mapping each point $\widehat{p} \in \widehat{Y}_C$ to any point $p \in X_C$ within the pixel represented by $\widehat{p}$.

**Universal-Baker.** We now give a theoretical instantiation of NN-Baker using the neural network obtained by the following universal approximation result.

**Theorem 3.1** (Cybenko [22]). *Let $\sigma$ be any continuous sigmoidal function. Let $m \in \mathbb{N}$, and let $\mathcal{C}$ be a compact subset of $\mathbb{R}^m$. Let $f : \mathcal{C} \to \mathbb{R}$ be a continuous function, and $\gamma > 0$. Then, there exists $N \in \mathbb{N}$, $a_1, \ldots, a_N \in \mathbb{R}$, $y_1, \ldots, y_N \in \mathbb{R}^m$, and $\theta_1, \ldots, \theta_N \in \mathbb{R}$, such that the function $g : \mathcal{C} \to \mathbb{R}$, with $g(x) = \sum_{i=1}^N a_i \sigma(y_i^T x + \theta_i)$, satisfies $\sup_{x \in \mathcal{C}} |g(x) - f(x)| < \gamma$.*

Using the neural network given by the above result, we can then argue that our NN-Baker has the capacity (expressiveness) of solving MIS problem (for unit-ball graphs) in a bi-criteria approximation. The simple proof of this theorem can be found in the Supplement.

**Theorem 3.2.** *Let $\varepsilon, \delta > 0$. There exists $N = N(\varepsilon, \delta, d)$, such that the following holds. Suppose that the function $g_{\mathcal{N}^*}$ in Step 2' of the NN-Baker framework is computed by the neural network $\mathcal{N}^*$ given by Theorem 3.1, with a single hidden layer of size $N$. Then, the resulting algorithm is a $(1 + \Theta(\varepsilon), 1 + \Theta(\delta))$-bi-criteria approximation in expectation for $d$-MIS, and it will call this (same) neural network at most $n$ times where $n$ is the number of points generating the input graph.*

**Remark 3.1.** *A similar statement to Theorem 3.2 holds for the Vertex Cover problem, yielding a $(1 + \Theta(\varepsilon), 1 + \Theta(\delta))$-bi-criteria approximation using the modifications discussed in Section 2.3.*

### 3.2 Instantiation of NN-Baker

Above we introduce a generic NN-Baker framework and a theoretical instantiation. We now provide two specific practical instantiations of this framework: a CNN-Baker and a GNN-Baker where (Step 2') of NN-Baker (or equivalently, (Step 2) of our Baker-paradigm) is implemented by a CNN or by a specific GNN. We provide details below. Specifically, recall that the input is a set of points $X \subset \mathbb{R}^d$, and for a randomly shifted grid $\Gamma + \tau$, let us focus on a specific grid cell $C \in \Gamma + \tau$. Recall that in NN-Baker, given C and $\widehat{X}_C$, we will use a neural network to compute a subset of $\widehat{Y}_C$ which ideally approximates a maximum independent set of the (unit-ball) geometric intersection graph $G_{\widehat{X}_C}$ spanned by $\widehat{X}_C$. The set $\widehat{Y}_C$ (of pixels) is further relaxed to a subset $Y_C \subset X_C$ as an independent set solution within cell C.

**CNN-Baker.** We can view the set of pixels in a cell as an image of size $\beta^d = \beta \times \beta \ldots \beta$ where $\beta = \Theta(\frac{1}{\varepsilon\delta})$. Specifically, given a grid cell C and a subset of pixels $Z$, we define the $I_Z$ over $\beta^d$ such that $I_Z[\mathsf{p}] = 1$ if $\mathsf{p} \in Z$ and $I_Z[\mathsf{p}] = 0$ otherwise. We call such an image *an induced image by* C, and denote the space of images of size by $SpIm$. Now, in (Step 2') of NN-Baker, we are given a subset of pixels $(Z =)\widehat{X}_C$ within a cell C, and aim to compute a subset $\widehat{Y}_C \subseteq \widehat{X}_C$ that forms an independent set for the unit-ball graph spanned by $Z = \widehat{X}_C$. Our CNN-Baker uses a CNN architecture $\mathcal{N}_C$ to compute a map $g_{\mathcal{N}_C} : SpIm \to SpIm$, which takes as input $I_{\widehat{X}_C}$ and outputs $I_{\widehat{Y}_C}$. Since the input and output are both images of the same size, this problem can be also viewed as an image

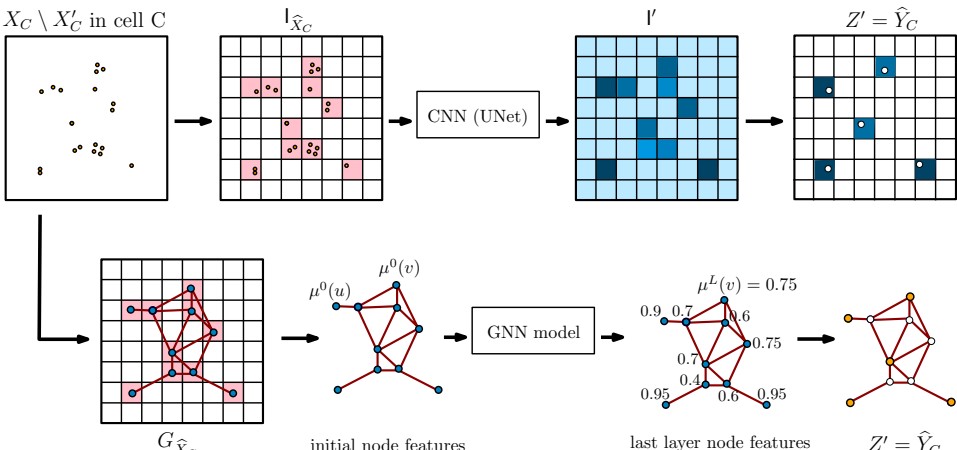

Figure 2: Given the set of points $X_C \setminus X'_C$ contained in cell $C$, the top row shows the processing of it by a CNN component as in CNN-Baker, while the bottom row shows it for a GNN component as in GNN-Baker. $\widehat{Y}_C$ in the figure corresponds to the set of pixels containing points from $Y_C$.

segmentation problem, and thus we can use a UNet architecture for the map $\mathcal{N}_C$. More specifically, the input is a binary image $I_{\widehat{X}_C}$, and the output is a grayscaled image $I'$ where the value at each pixel, $I'[i] \in [0,1]$, indicates the likelihood that this pixel belongs to a maximum independent set in the geometric intersection graph $G_{\widehat{X}_C}$.

In (Step 2') of NN-Baker, we describe an alternative greedy approach to convert this likelihood map to a subset of points $Y_C \subset X_C \setminus X'_C$ as the output MIS. However, in our implementation for both CNN-Baker and GNN-Baker, we will make a slight modification: In particular, note that the output from (Step 2') will be a $(1 + \Theta(\delta))$-independent subset of $X_C$. In practice, we would like to guarantee that we output a valid MIS for $X_C$ (i.e, any two points in our output $Y_C$ should be at least distance 2 apart). We thus follow the greedy approach as outlined in (Step 2') but with a small modification: We sort all pixels in $Z = \widehat{X}_C$ in decreasing order of their pixel values, and assume $Z = \{z_1, \ldots, z_\ell\}$ is this sorted list. We then inspect them one-by-one in order. At the beginning, initialize an output set $Y'$ to be empty. Then in the $i$-th iteration, let $y_i$ be any point from $X_C \setminus X'_C$ in pixel $z_i$. If $y_i$ is independent to all points in $Y'$ (i.e, it is more than 2 away for all points in $Y'$), add $y_i$ to $Y'$; and otherwise, do nothing. In the end after $\ell$ iterations, we obtain a set of $Y' \subseteq X_C \setminus X'_C$ points (not pixels) which is guaranteed to be an independent set for the unit-ball graph spanned by points in $X_C$. Set $Y_C = Y'$ and return it as the independent set for this call $C$. This will further guarantee that the final output $\bigcup_C Y_C$ computed by our CNN-Baker will be a valid MIS for the unit-ball graph spanned by input points $X$.

**GNN-Baker.** For GNN-Baker, we instead directly use the unit-ball graph $G_{\widehat{X}_C}$ as input, with the initial node feature $\mu^0(v)$. At the last $L$-th layer, the node feature $\mu^L(v)$ gives the likelihood that $v$ is in the maximum independent set. We then retrieve an independent set $Y_C$ by the same greedy approach as for the case of CNN-Baker. For the specific choice of the GNN architecture, we can use any of the existing models, such as GCN [37], GraphSAGE [38], GIN [39], GAT [40], and VV$_C$-GNN [13]. In our later experiments, we will use TGS [12] and LwD [16] in our GNN-Baker framework, as these are state-of-the-art (SOA) approaches specifically designed for graph combinatorial problems. (Our experiments show that using a general purpose GNN has much worse performance than TGS and LwD.) In particular, TGS, a supervised learning approach, takes GCN to process reduced graphs and use a tree search approach to label nodes whether they are in an independent set or not. LwD is reinforcement-learning based, and designs a policy network and value network on each MIS problem state with GraphSAGE architecture. We will also use the Erdős-GNN [15], designed for optimization problems in the *unsupervised setting*. Erdős-GNN takes multiple GIN layers followed by a GAT layer to learn graphs' distribution. It designs a differentiable loss function based on expectations of optimization problems objective functions and a probabilistic penalty function.

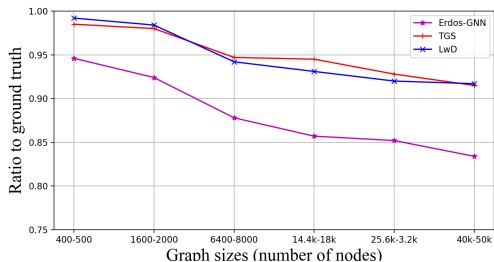

Figure 3: Results of Erdos-GNN, TGS and LwD on MIS problems with different sizes of graphs in 2D setting. We report the ratio to ground truth computed by KaMIS.

# 4 Experimental results

We present results for $d$-MIS here. More results, including for minimum vertex cover (MVS), can be found in the Supplement. We consider 5 families of unit-ball graphs: (2D-dense) consists of a set of graphs, each with points distributed uniformly on a 2D rectangular region with around 40K – 50K points. (2D-sparse) consists of a set of graphs, each with 40k – 50k points distributed uniformly on a 2D rectangular region four times larger than the dense region . (2D-Gaussian) consists of graphs each spanned by points sampled from a Gaussian-mixture distribution with 40K – 50K points over the same region size as the dense region. (3D) consists of graphs each spanned by points sampled from a 3D region. (Torus-4D) consists of graphs each spanned by points sampled from a torus embedded in $\mathbb{R}^4$ (see Supplement for details).

First, we note both CNN-Baker and GNN-Baker return valid MIS for the input unit-ball intersection graphs (as detailed in the implementation of CNN-Baker). To report accuracy, we will need ground truth solutions. However, computing exact solutions for all our test cases is computationally intractable. Instead, we use the output of a SOA solver KaMIS [41] as the ground truth solutions, and report the average ratio of MIS obtained over this ground-truth MIS sizes as a metric for accuracy (so the larger this ratio is the better). To validate the accuracy of KaMIS on our training and test data, we compared against an exact solution [42] for 20 test sets and found that the accuracy of KaMIS exceeded 99.9% of the optimal solution in all cases. On the other hand, regardless the accuracy of KaMIS, since we are taking the ratio as the metric for accuracy, higher ratio is always better.

Before we show results of CNN-Baker and GNN-Baker, we first show how the SOA GNN-based approaches, TGS [12], LwD [16] and Erdős-GNN [15], all of which are specifically designed for graph optimization problems (see discussions in Section 3.2), perform as the size of graph increases. Here for each target size, 1000 graphs spanned by points sampled from the same dense distribution in 2D are used for training, then tested on 100 graphs of roughly the same target size. As shown in Figure 3, the accuracy decreases as the size of (geometric) graphs increase.

Table 1: The ratio of MIS results from different GNNs, GNN-Baker approaches and K-Baker approach to ground truth. (Larger values are better.)

|  | UNetBaker | Erdős | ErdősBaker | TGS | TGSBaker | LwD | LwDBaker |
|---|---|---|---|---|---|---|---|
| 2D-dense | 0.915 | 0.834 | 0.923 | 0.915 | 0.936 | 0.917 | **0.955** |
| 2D-sparse | 0.919 | 0.822 | 0.917 | 0.909 | **0.931** | 0.908 | 0.926 |
| 2DGaussian | 0.917 | 0.769 | 0.848 | 0.905 | **0.927** | 0.911 | 0.925 |
| 3D | - | 0.856 | 0.930 | 0.924 | 0.948 | 0.902 | **0.954** |
| Torus-4D | - | 0.812 | 0.926 | 0.923 | **0.937** | 0.910 | **0.937** |

**NN-Baker setup.** For each of the five setups, we train on 1000 graphs and test on 200 graphs each containing between 40k - 50k points. However, inside our NN-Baker framework, the input domain is partitioned into cells of side-length 12.8, and each cell is further partitioned into $128 \times 128$ pixels (each with side-length $= 0.1$). In other words, each cell can be viewed a 128x128 image. This means that the training set for the NN component involved contains only small graphs restricted to such cells. In the end, each training graph consists of around 400 points for the cases of (2D-dense), (3D) and (Torus-4D), and around 100 to 125 points for the case of (2D-sparse). The size of each small

training graph for Gaussian case is non-uniform. For CNN-Baker, we apply a UNet which reduces the input 128x128 image to an 8x8 image after 4 down-scaling layers, each of which consists of two convolutional layers, a dropout layer and a max pooling layer. From the 8x8 image, there are then 4 upsampling/concatenation layers to bring the size back to 128x128. This model is denoted by UNet-Baker in Table 1. For GNN-Baker, as mentioned in Section 3.2, we test TGS-Baker, LwD-Baker, and Erdős-Baker. For each individual neural network involved, we use the same training setup and hyperparameters as those in their official implementations. For LwD and Erdős-GNN, we use node degree as initial node features, while for TGS, we simply use a constant vector. The accuracy of these different methods over the 5 families of graphs are shown in Table 1. The number of parameters for these GNN-Bakers range from 50K to 600K, while the UNet-Baker uses around 80M parameters.

As shown in Table 1, using Baker framework consistently improves the performance of these SOA neural networks on geometric MIS problems. The improvement over the unsupervised Erdős-GNN (i.e, Erdős-Baker vs. Erdős-GNN) is particularly significant. We remark that we have also experimented with using a simple multi-layer fully connected NN (i.e., a multi-layer version of the NN used in Theorem 3.2) to instantiate our NN-Baker, and the performance (under similar number of parameters as our UNet-Baker) is much worse, between 70%-80%. We can also use NN-Baker trained on one type of graphs (e.g, 2D-dense) but apply it to graphs from the other families to study the generalization of the resulting framework – we observe that the accuracy decreases but still improves over the non-Baker version. See the Supplement for these more results, as well as the performance of NN-Baker for minimum vertex cover problem.

**Timing.** To show that the NN-Baker is efficient compared to a traditional Baker paradigm, we compare the runtime of our models against a non-neural network based approach. For this, we use KaMIS as our solver for each cell, and call the resulting framework **K-Baker**. In Table 2, we show the average time taken to solve a problem with 40k-50k points. We show the runtime of K-Baker set to achieve similar performances to UNetBaker in the table. If we set a similar runtime to UNetBaker, then K-Baker's performances on 2D-dense, 2D-sparse and 2DGaussian are 0.753, 0.672 and 0.674, which are much poorer than our NN-Baker.

Table 2: Average solve times of KaMIS, NN-Baker and K-Baker (seconds)

|  | KaMIS | K-Baker | UNetBaker | ErdősBaker | TGSBaker | LwDBaker |
|---|---|---|---|---|---|---|
| 2D-dense | 3385.1 | 415.19 | 18.06 | 32.36 | 15.72 | 44.55 |
| 2D-sparse | 2468.8 | 402.57 | 59.97 | 68.95 | 33.74 | 81.06 |
| 2DGaussian | 3166.7 | 420.83 | 58.05 | 47.45 | 28.38 | 62.59 |

## 5 Concluding remarks

The advancement in neural network architectures and their potential to adapt to the structure and input distribution of a problem in a data-driven manner, have brought new ways to tackle traditionally challenging tasks, such as graph optimization problems. In this paper, we advocate two points of view: (1) Problems in geometric settings can provide structures that both algorithms and neural networks can leverage; for example, they can help to decompose problems into local versions of bounded size and thus lead to more effective NN components. (2) Infusing NN + learning into an algorithmic paradigm can lead to a more powerful framework for hard problems, potentially with theoretical guarantees. While the latter is a view that has already attracted momentum in recent years, our work provides new perspectives (e.g, the decomposition into bounded-size sub-problems) together with some theoretical guarantees, and we show that the resulting method is indeed more powerful empirically too. Our present algorithms currently apply to only geometric intersection graphs. Nevertheless, we believe that such ideas go beyond the geometric setting which we hope to explore in the future, such as to frameworks to obtain algorithms for graphs with bounded tree-width. Indeed, the algorithms and theoretical computer science community has developed many beautiful algorithmic paradigms that may be suitable to be infused with NN+ML ideas. Finally, to our best knowledge, this work does not have direct negative societal impacts.

**Acknowledgement.** This work is in part supported by the National Science Foundation (NSF) under grants CCF-1815145, IIS-1815697, and CCF-2112665.

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
