# Supplement: NN-Baker: A Neural-network Infused Algorithmic Framework for Optimization Problems on Geometric Intersection Graphs

**Evan McCarty** [*]
Department of Computer Science
University of Illinois, Chicago
emccarty@uic.edu

**Qi Zhao** [*]
Computer Science and Engineering Department
University of California, San Diego
qiz032@ucsd.edu

**Anastasios Sidiropoulos**
Department of Computer Science
University of Illinois, Chicago
sidiropo@uic.edu

**Yusu Wang**
Halıcıoğlu Data Science Institute
University of California, San Diego
yusuwang@ucsd.edu

## 1 Missing proofs

### 1.1 Proof of Theorem 2.1

In what follows, we use the notation $[k]$ to denote the set of integers $1, 2, \ldots, k$. Recall that each cell of our axis-parallel hyper-grid $\Gamma$ has side-length $\frac{2d}{\varepsilon}$. Also recall that we put on a second-level more refined lattice (grid) which gives us pixels of side-length $\delta' = \delta/\sqrt{d}$. For simplicity of argument, we assume that $\frac{1}{\delta'}$ (i.e, the reciprocal of the pixel side-length) is an integer. This condition can be removed by slightly more careful analysis. In what follows, we will show that the output of Baker's paradigm, $Y$, is a $(1 + 2\varepsilon, 1 + 6\delta)$-bi-criteria approximation in expectation for $d$-MIS for $0 < \varepsilon, \delta < 1/3$, with the desired time complexity.

Now fix some optimal solution $Y^* \subseteq X$ for $d$-MIS over $X$. Let $Y' \subseteq Y^*$ be the subset of points in $Y^*$ such that the unit balls centered at them intersect the shifted grid $\Gamma + \tau$; that is

$$Y' = \{p \in Y^* : \mathsf{ball}(p, 1) \cap (\Gamma + \tau) \neq \emptyset\}.$$

Note that $Y' \subseteq X'$ as constructed in (Step 2) of the main paper. For any point $p \in X$, for any $i \in [d]$, let $\mathcal{E}_{p,i}$ be the event that $\mathsf{ball}(p, 1)$ intersects some $(d-1)$-dimensional hyperplane of the shifted grid $\Gamma + \tau$, that is orthogonal to $e_i$. The event $\mathcal{E}_{p,i}$ occurs precisely when the $i$-th coordinate of $\tau$ falls within an interval of length 2 out of the side-length $\frac{2d}{\varepsilon}$ of a cell: This is because it is equivalent to that we have a segment of length $2d/\varepsilon$ (along the $i$-th axis in direction $e_i$), and a point is within distance 1 to either endpoint of this segment. Hence the total probability is the same as a point to fall within an interval of length $1 + 1 = 2$ out of an interval of length $2d/\varepsilon$. Since $\tau$ is chosen uniformly at random, we get

$$\Pr[\mathcal{E}_{p,i}] = \frac{2}{2d/\varepsilon} = \varepsilon/d.$$

Let $\mathcal{E}_p$ be the event $p \in Y'$. By the union bound, we have

$$\Pr[\mathcal{E}_p] = \Pr[\bigcup_{i \in [d]} \mathcal{E}_{p,i}] \leq \sum_{i \in [d]} \Pr[\mathcal{E}_{p,i}] = \varepsilon.$$

---

[*]Equal contribution

35th Conference on Neural Information Processing Systems (NeurIPS 2021).

Using linearity of expectation, the above implies

$$\mathbf{E}[|Y'|] = \sum_{p \in Y^*} \Pr[\mathcal{E}_p] \le |Y^*| \cdot \varepsilon = \mathsf{OPT}(X) \cdot \varepsilon. \tag{1}$$

In other words, if we only consider points in $X \setminus X'$, then the size of the optimal solution of $d$-MIS for $X \setminus X'$ can only be at most $\mathsf{OPT}(X) \cdot \varepsilon$ less than $\mathsf{OPT}(X)$, that is, it is at least $(1 - \varepsilon)\mathsf{OPT}(X)$.

Recall that $\widehat{X}$ is the "snapping" of of points in $X \setminus X'$ to the second-level lattice points: in particular, a point $p = (p_1, \ldots, p_d) \in X \setminus X'$ is mapped to the point $\widehat{p} = (\delta' \lfloor \frac{p_1}{\delta'} \rfloor, \ldots, \delta' \lfloor \frac{p_d}{\delta'} \rfloor)$ which intuitively is the ($d$-dimensional analog of the) left-bottom of the pixel (of side length $\delta'$) in the second-level lattice containing $p$. Let $G_{\widehat{X}}$ be the intersection graph spanned by balls centered points in $\widehat{X}$ but with radius $1 - \delta$. First, note that as all points within 1 from cell-boundaries are removed[2], we have that each connected component of $G_{\widehat{X}}$ has to be contained inside some cell $\mathsf{C}$ of $\Gamma + \tau$. Hence to compute MIS (maximum independent set) for $G_{\widehat{X}}$, we can do so by computing an optimal MIS for $G_{\widehat{X}_\mathsf{C}}$, the restriction of $G_{\widehat{X}}$ within every cell $\mathsf{C}$ of $\Gamma + \tau$, and then the union of them over all cells is necessarily an MIS for $G_{\widehat{X}}$. In (Step 2), we compute $\widehat{Y}_\mathsf{C}$, which is an MIS for $G_{\widehat{X}_\mathsf{C}}$. If we take the union of this for all cells, namely $\widehat{Y} = \bigcup_\mathsf{C} \widehat{Y}_\mathsf{C}$, then it is clear that $\widehat{Y}$ is an MIS for $G_{\widehat{X}}$.

Furthermore, for any cell $\mathsf{C}$, as it is a $d$-dimensional hypercube of side-length $2d/\varepsilon$, we have that its volume is $(2d/\varepsilon)^d$. As any independent set in the cell necessarily has pairwise distance $> 2$, it follows that the maximum cardinality of any independent set in $C$ is at most $s = (2d/\varepsilon)^d/V_d = V_d^{-1}(2d/\varepsilon)^d$, where $V_d$ stands for the volume of a radiue $(1 - \delta)$ ball in $\mathbb{R}^d$. Furthermore, there are only $M := (\frac{2d}{\varepsilon\delta'})^d = (\frac{2d\sqrt{d}}{\varepsilon\delta})^d$ number of pixels inside cell $\mathsf{C}$, we can thus enumerate all possible independent sets for $\widehat{X}_\mathsf{C}$ in time $M^s = (\frac{1}{\varepsilon\delta})^{(d/\varepsilon)^{O(d)}}$ time. Since there are at most $n$ cells of $\Gamma + \tau$ that contains non-empty $\widehat{X}_\mathsf{C}$, the total time to construct an MIS for $G_{\widehat{X}}$ is thus $(\frac{1}{\varepsilon\delta})^{(d/\varepsilon)^{O(d)}} n$ as claimed in Theorem 2.1.

Note that in (Step 2) of Baker's paradigm, after computing $\widehat{Y}_\mathsf{C}$, we need to transfer it to a subset $Y_\mathsf{C} \subseteq X_\mathsf{C} \subseteq X$ of original input points. In particular, we achieve this by mapping each point $\widehat{p} \in \widehat{Y}_\mathsf{C}$ to an arbitrary point $p = \pi(\widehat{p}) \in X_\mathsf{C}$ contained in the pixel that $\widehat{p}$ is the bottom-left corner of (this is a consequence of the construction of set $\widehat{X}_\mathsf{C}$, where we snap each point $q$ in $X_\mathsf{C}$ to the left-bottom vertex of the pixel $q$ lies in). Obviously, this map $\pi : \widehat{Y}_\mathsf{C} \to Y_\mathsf{C}$ is a bijection, and the distance $\|\widehat{p} - \pi(\widehat{p})\|_2 \le \delta$. (Note that the diameter of a pixel in the cell $\mathsf{C}$ is $\delta$ as the side-length of this pixel is $\delta' = \delta/\sqrt{d}$.)

What remains is to prove that $Y = \bigcup_\mathsf{C} Y_\mathsf{C}$ as computed in (Step 3) of Baker's paradigm is indeed a bi-criteria approximation in expectation for the MIS of $G_X$, the unit-ball intersection graph spanned by input points $X$.

To this end, first, note that $\widehat{Y} = \bigcup_\mathsf{C} \widehat{Y}_\mathsf{C}$ is a maximum independent set for $G_{\widehat{X}}$ as argued earlier. We claim that $|\widehat{Y}| \ge |Y^* \setminus Y'|$. This is because that since $Y^* \setminus Y'$ is an independent set for the unit-ball graph spanned by points in $X \setminus X'$, we have that for any points $y, y' \in Y^* \setminus Y'$, $\|y - y'\|_2 > 2$. Now map $y$ and $y'$ to $\hat{y}$ and $\hat{y}'$, the respective left-bottom corner of the pixels they are contained in; note that $\hat{y}, \hat{y}' \in \widehat{X}$. By the triangle inequality, we have that $\|\hat{y} - \hat{y}'\|_2 \ge 2 - 2\delta$ as the diameter of each pixel is $\delta$. This means that snapping all points in $Y^* \setminus Y'$ as such to points in $\widehat{X}$ gives rise to an independent set of $G_{\widehat{X}}$. This in turn implies that a maximum independent set of $G_{\widehat{X}}$, namely $\widehat{Y}$, is at least as large as the set $Y^* \setminus Y'$; that is, $|\widehat{Y}| \ge |Y^* \setminus Y'|$. Combining this with Eqn (1), we then have that

$$\mathbf{E}[|Y|] = \mathbf{E}[|\widehat{Y}|] \ge |Y^*| - \mathbf{E}[|Y'|] \ge (1 - \varepsilon)|Y^*| = (1 - \varepsilon)\mathsf{OPT}(X) \ge \mathsf{OPT}/(1 + 2\varepsilon), \tag{2}$$

where the last inequality holds for any positive $\varepsilon < 1$. This establishes one side (lower-bound side) of the bi-criteria approximation.

We now consider the upper-bound in the bi-criteria approximation. Note that we have a bijection $\pi : \widehat{Y} \to Y$ which sends a point $\widehat{p} \in \widehat{Y}$ to a point $\pi(p)$ within $\delta$ distance. Combining this with the

---

[2]We removed all points in $X$ within distance 1 from the cell boundaries. But since we assume that the $1/\delta'$, the reciprocal of the pixel side-length, is an integer, this statement about points in $\widehat{X}$ still holds.

fact that $\widehat{Y}$ itself is an independent set for $G_{\widehat{X}}$ (i.e, any two points inside are at least distance $2 - 2\delta$ apart), we have that $Y$ is an $(1 + 6\delta)$-independent set: This is because any two points in $Y$ are at least $2 - 4\delta \geq 2/(1 + 6\delta)$ apart, as $1 - 2\delta \geq 1/(1 + 6\delta)$ holds for any positive $\delta < 1/3$. It then follows that $|Y| \leq \mathsf{OPT}_{1+\Theta(\delta)}(X)$.

Putting both sides (upper and lower bounds) together, we have that the set $Y$ computed by our proposed Baker's paradigm is a $(1 + \Theta(\varepsilon), 1 + \Theta(\delta))$-bicriteria approximation of $d$-MIS for input point set $X$ in expectation. Together with the time complexity bound computed earlier, this concludes the proof of Theorem 2.1.

## 1.2 Removing the bi-criteria condition and Theorem 2.2

We note that one can easily modify our algorithm Baker-MIS to obtain a $(1 + \Theta(\varepsilon))$-approximation for MIS (instead of a bi-criteria approximation), by trading off a slower running time, similarly to [1]. We include the details here for completeness. The algorithm proceeds exactly as Baker-MIS, with the only difference being that Step 2 is replaced by the following:

> **Step 2'': Solving the problem exactly locally on each cell.** For each cell $\mathsf{C}$ of $\Gamma + \tau$, let $X_\mathsf{C}$ be the restriction of $X \setminus X'$ to cell $\mathsf{C}$. Now in this modified step $2''$, we will work with $X_\mathsf{C}$ instead of working with the set $\widehat{X}_\mathsf{C}$, which is the snapping of set $X_\mathsf{C}$ to pixels in the cell. Let $G_{X_\mathsf{C}}$ be the intersection graph of unit balls centered at the points in $X_\mathsf{C}$; that is $V(G_{X_\mathsf{C}}) = X_\mathsf{C}$, and
>
> $$E(G_{X_\mathsf{C}}) = \left\{ \{p, q\} \in \binom{X_\mathsf{C}}{2} : \|p - q\|_2 \leq 2 \right\}.$$
>
> We compute the maximum independent set $Y_\mathsf{C}$ in $G_{X_\mathsf{C}}$ which we know has size at most $s = V_d^{-1}(2d/\varepsilon)^d$, where $V_d$ denotes the volume of the $d$-dimensional unit ball. This can be done by enumerating all possible subsets of $X_\mathsf{C} = V(G_{X_\mathsf{C}})$ of size at most $s$, and taking the maximum cardinality such subset that is independent in $G_\mathsf{C}$.

In (Step 3), we will return $Y = \bigcup_\mathsf{C} Y_\mathsf{C}$ as before. As seen in Theorem 2.2, the price to pay to obtain a standard $(1 + \varepsilon)$-approximation is that the dependency of time complexity on $n$ increases from previous $n$ (i.e, linear) to $n^{(1/\varepsilon)^{O(d)}}$. This is because during the exhaustive enumeration to solve MIS for $G_{X_\mathsf{C}}$, we have to take all subsets of $X_\mathsf{C}$ of size at most $s$. Since the cardinality of $X_\mathsf{C}$ could be $n$ (say when all points in $X$ happen to be inside a single cell $\mathsf{C}$ of the randomly shifted grid $\Gamma + \tau$), we thus needs $n^{(1/\varepsilon)^{O(d)}}$ time for this enumeration. The approximation guarantee follows the proof of Theorem 2.1, but as $Y$ constructed is now a valid independent set for $X$, we do not have the relaxation of $(1 + \Theta(\delta))$-independent set. Theorem 2.2 thus follows.

## 1.3 Proof of Theorem 3.2

By Theorem 3.1 stated in the main text, we can obtain a neural network $\mathcal{N}^*$ with a single hidden layer that computes a function $g_{\mathcal{N}^*} : [0, 1]^k \to [0, 1]^k$, such that

$$\sup_{x \in [0,1]^k} |g_{\mathcal{N}^*}(x) - f_{\mathrm{MIS}}(x)| < 1/2,$$

where the hidden layer has size $N = N(\varepsilon, \delta, d)$. By rounding the output of $g_{\mathcal{N}^*}$, we obtain the indicator vector of a maximum-independent set $\widehat{Y}_\mathsf{C}$ for $\widehat{X}_\mathsf{C}$. The same holds for the greedy strategy to choose an output as described in (Step $2'$) of NN-Baker. The proof of Theorme 2.1 states that Eqn (2) holds for $\widehat{Y} = \bigcup_\mathsf{C} \widehat{Y}_\mathsf{C}$.

Next, we map $\widehat{Y}_\mathsf{C}$ to $Y_\mathsf{C}$ as before by mapping each $\widehat{p} \in \widehat{Y}_\mathsf{C}$ to $\pi(p)$ on $X_\mathsf{C}$ within $\delta$ distance to $\widehat{p}$. Following the same argument as in the proof of Theorem 2.1, we know that the resulting $Y = \bigcup_\mathsf{C} Y_\mathsf{C}$ is a $(1 + \Theta(\varepsilon), 1 + \Theta(\delta))$-bicriteria approximation in expectation of $d$-MIS for the input points $X$. Finally, since there will be at most $n$ cells containing at least one point from $X$, we know that we only need to call this neural network $\mathcal{N}^*$ at most $n$ times. This completes the proof of Theorem 3.2.

## 2 Additional experimental results

**Hardware information**  All baselines and NN-Baker models are trained and test on an AMD-EPYC-7452 CPU and a RTX-A6000 GPU.

**Additional dataset information**  2D-Gaussian dataset is a collection of geometric graphs generated from 40k - 50k points sampled from a 2D mixture of 5 Gaussian distribution. The centers of this 5 Gaussian distribution are $[(64, 64), (32, 32), (32, 96), (96, 32), (96, 96)]$ and their standard deviance is 20.0. The input domain is partitioned into cells of side-length 12.8, and each cell is further partitioned into $128 \times 128$ pixels. 3D dataset is a collection of geometric graphs generated by points uniformly sampled from a 3D cube region with around 40k - 50k points. Each cell in the domain has side-length 5, and is partitioned into $50 \times 50 \times 50$ pixels. Torus-4D dataset is a collection of geometric graphs generated by points sampled from a 4D surface with around 40k - 50k points. The 4D surface is generated by two functions $f, g : [0, 1]^2 \rightarrow \mathbb{R}^4$ that:

$$\begin{aligned}
f(\alpha) &= (r \sin \alpha, r \cos \alpha, 0, 0) \\
g(\beta) &= (0, 0, r \sin \beta, r \cos \beta)
\end{aligned} \tag{3}$$

where $r > 0$ is a constant, and we set $r = 20$ in experiments. Given a point set $X$ uniformly sampled from $[0, 1]^2$, we have a resulting point set $(f + g)(X) \subset \mathbb{R}^4$. Each cell in the 4D surface is mapped from a cell with side-length 0.1 in $[0, 1]^2$ which is partitioned into $100 \times 100$ pixels.

**More experimental statistics**  As a baseline comparison, we compared our CNN and GNN approaches to a standard feed forward NN. Both models were trained on the image segmentation problem of converting 128x128 images with the points as inputs to 128x128 images of points which should be in the independent set. This was combined with the Baker technique to produce the figures in Table 1. For the "Small" neural network, this model contained two hidden layers and a total number of parameters equal to our UNet-Baker approach (76 million). The "Large" model also contains two hidden layers and a total number of parameters roughly equal to double that value (147 million). For this data, models were trained and tested on data from the same distribution.

Table 1: Performance on MIS by fully connected models

|  | Small | Large |
|---|---|---|
| 2D-dense | 0.714 | 0.788 |
| 2D-sparse | 0.789 | 0.898 |
| 2DGaussian | 0.724 | 0.852 |

To report the variance of the architectures proposed, we trained ten models for each architecture from different random start weights. We report the standard deviations ($\times 10^{-3}$) of the ratios of MIS results from our models to ground truth in Table 2.

Table 2: The standard deviations of MIS results from different models ($\times 10^{-3}$).

|  | UNetBaker | Erdős | ErdősBaker | TGS | TGSBaker | LwD | LwDBaker |
|---|---|---|---|---|---|---|---|
| 2D-dense | 3.06 | 2.70 | 5.83 | 1.15 | 3.72 | 3.58 | 6.75 |
| 2D-sparse | 2.93 | 2.73 | 8.25 | 1.12 | 3.49 | 3.61 | 8.53 |
| 2DGaussian | 3.10 | 10.62 | 14.36 | 2.35 | 4.28 | 8.57 | 7.82 |
| 3D | - | 3.53 | 5.85 | 1.82 | 2.95 | 5.42 | 8.63 |
| Torus-4D | - | 5.64 | 5.24 | 2.10 | 3.87 | 5.35 | 8.30 |

To show the generalization power of the NN-Baker models, we tested our models on data with different distributions than the data was trained. For these, we used the data same three 2D distributions as our other results. We report the ratio of MIS results from these generalized models to ground truth in Table 3.

Table 3: The ratio of MIS results from generalized models to ground truth

| Train | Test | UNetBaker | ErdősBaker | TGSBaker | LwDBaker |
|---|---|---|---|---|---|
| 2D-dense | 2D-sparse | 0.910 | 0.901 | 0.925 | 0.914 |
|  | 2DGaussian | 0.912 | 0.832 | 0.912 | 0.915 |
| 2D-sparse | 2D-dense | 0.915 | 0.908 | 0.926 | 0.936 |
|  | 2DGaussian | 0.917 | 0.825 | 0.919 | 0.908 |

We also evaluate the performance of CNN-Baker and GNN-Baker on solving minimum vertex cover (MVC) problems. The training and test dataset is the same as what we take in MIS problems, and the ground truth is computed by KaMIS. We report the ratio of MVC results from different approaches to ground truth in Table 4. Since this is a minimization problem, results closer to 1.0 are more optimal.

Table 4: The ratio of MVC results from different approaches to ground truth.

|  | UNetBaker | Erdős | ErdősBaker | TGS | TGSBaker | LwD | LwDBaker |
|---|---|---|---|---|---|---|---|
| 2D-dense | 1.210 | 1.141 | 1.066 | 1.072 | 1.054 | 1.071 | 1.038 |
| 2D-sparse | 1.301 | 1.531 | 1.248 | 1.271 | 1.206 | 1.274 | 1.221 |
| 2DGaussian | 1.234 | 1.203 | 1.133 | 1.084 | 1.064 | 1.078 | 1.066 |

Post-processing is an important part of our implementations. After each of the cells are solved by the NN-Baker framework, we then add in points close to the boundaries that do not intersect any of the points already in the set. The percentage of points added for all methods is given in Table 5.

Table 5: Proportion of points added in post processing

|  | UNetBaker | ErdősBaker | TGSBaker | LwDBaker |
|---|---|---|---|---|
| 2D-dense | 0.085 | 0.080 | 0.025 | 0.032 |
| 2D-sparse | 0.086 | 0.043 | 0.005 | 0.006 |
| 2DGaussian | 0.082 | 0.042 | 0.025 | 0.016 |

## References

[1] Dorit S Hochbaum and Wolfgang Maass. Approximation schemes for covering and packing problems in image processing and vlsi. *Journal of the ACM (JACM)*, 32(1):130–136, 1985.