# OpenReview forum: "NN-Baker: A Neural-network Infused Algorithmic Framework for Optimization Problems on Geometric Intersection Graphs"
_NeurIPS.cc/2021/Conference — NeurIPS 2021 Poster_

### Official Review · Reviewer_88CS · 2021-07-08

**Rating:** 7
**Confidence:** 4

**Summary:**

This paper studies graph optimization problems in the geometric domain and proposes a novel paradigm of neural combinatorial optimization, NN-Baker. NN-Baker breaks down the original problem into local subproblems of bounded sizes. Neural networks can solve the subproblems with theoretical guarantees thanks to the universal approximation theorem. The proposed method performs better than vanilla methods in several synthetic datasets.

**Limitations And Societal Impact:**

They are discussed in Section 5.

**Main Review:**

# Strengths

* The proposed framework, NN-Baker, is motivated by results from theoretical computer science and seems quite novel in the machine learning literature. It provides the machine learning community with some insights.
* While the method is principled from the theoretical perspective, the resulting strategy is simple and intuitive.
* Thanks to the partitioning and coarsening mechanism, the subproblems neural networks actually need to learn are small, which leads a good training strategy for large combinatorial tasks.
* Figures 1 and 2 describe the method well.

# Weaknesses

* Task information is lost in three steps. First, peripheral points in each cell are ignored. Second, points are discretized into pixels. Third, the subproblems are solved by neural networks. I'm wondering about how much information is lost in each step. It would be interesting to see the performance when the subproblem in each cell is solved exactly.
* Only synthetic data are used. It would strengthen the paper if motivating real-world data could be used. City and sensor data can be candidates.

**Time Spent Reviewing:**

4

---

> ### Author Response · Authors · 2021-08-10
> **Response to Reviewer 88CS**
>
> We thank you for your time and all your comments! We are glad that you appreciate the novelty of the overall Baker-NN framework which integrates algorithmic insights with ML frameworks. We believe that the high-level idea of decomposing an input problem into small subproblems algorithmically and then using ML approaches to tackle these subproblems, is general and applicable to many problems beyond what’s currently studied in our submission (we have listed some in response to Reviewer YGgd, bullet point (2)).
>
> Regarding your comment that “Task information is lost in three steps. First, peripheral points in each cell are ignored. Second, points are discretized into pixels. Third, the subproblems are solved by neural networks. I'm wondering about how much information is lost in each step. It would be interesting to see the performance when the subproblem in each cell is solved exactly. ”
>
> Response:  Thank you for this comment. Indeed, information is lost in these three steps in our NN-Baker framework. For the first issue of peripheral points: as described in lines 211--215 of the submission, we bring back some peripheral points as long as they don’t form a violation to the independent condition (i.e, at least distance 2 to existing points in the independent set). All experimental results reported are after this enhancement. We show the effect of this step in the response to Reviewer JP57 (point (4) of the response): Roughly speaking, this post-processing improves results by between 0.5 -- 2.0% for GNN-Baker architectures, but it can improve 7% in performance for UNet-Baker architecture.
>
> Discretization is the second source of information loss. However, we expect this step to have minor effect in the output in practice -- An indirect evidence is in our response to Reviewer YGgd (point (4)) of the response), where by forcing output to be valid independent set without relaxing the distance threshold as in the bi-criteria approximation, the difference in the size of output independent-set changes negligibly.
>
> The third information loss is indeed interesting. Thank you for your suggestion, and we will include this ablation study (namely, compare with Baker+Exact solver) in the revised version of the paper.  In general, we cannot use an exact solver on all input test graphs as the running time will be prohibitively large. However, we will include a comparison with exact solutions for some cases where the branch-and-bound method terminates within reasonable time. We reported some new results in Baker+Heuristics, see response to Reviewer YGgd, point (1), that is relevant to your comment.

---

> > ### Comment · Reviewer_88CS · 2021-09-01
> > **Response**
> >
> > Thank you for the response.
> >
> > I read the response and other reviews. I think this paper contains good contributions. I maintain my original score and vote for acceptance.

---

> > > ### Author Response · Authors · 2021-09-01
> > > **Thanks**
> > >
> > > Thanks for your time and comments!

---

### Official Review · Reviewer_JP57 · 2021-07-16

**Rating:** 7
**Confidence:** 4

**Summary:**

The paper proposes a polynomial time approximation scheme that can utilize neural networks
to efficiently solve combinatorial problems like maximum independent set and minimum vertex cover
on geometric interesection graphs. The method decomposes a problem instance into multiple smaller
instances of bounded size located on a grid, which are then solved with the help of a neural network. The neural network
is either a graph neural network applied on the intersection graph or a CNN that treats the problem
as a segmentation task. To discretize the output of the network the authors use a greedy
rounding strategy that preserves feasibility.

**Limitations And Societal Impact:**

The authors have discussed the limitations of their work sufficiently.

**Main Review:**


### Strengths:
- The idea of reducing a problem to instances of bounded size in order to obtain an approximation guarantee in an ML setting is interesting.
- The use of neural networks (NN-Baker) seems to provide an improved theoretical result over the standard Baker approach, which is nice.
- Figures help build intuition and improve the readability of the paper considerably.
- Related work is sufficiently discussed. Experiments include models from recently published works.
- Seems to consistently improve the performance of models from the literature.
- Nice experiment on generalization.


### Criticism, comments, and questions:

- In terms of writing quality, there are plenty of typos and minor mistakes.
Here are some examples:

line 53: "Does a proposed ML pipeline has the expressiveness capacity..."

line 115: "Our NN Baker framework instead calls the neural network involved only linear time."

Text in Figure 1: "(Step 2) of Baker's paradigmfor each cell"

table 1 caption: "...approaches to groud truth"

line 217: "Detials ae in the supplement."


lines 231-232: is at most and \epsilon|OPT(X)|.

line 349: "For each of the five setup,"

line 366-367: "As shown in Table 1, using Baker framework consistently improve the performance of these SOA
neural networks on geometricMIS problem."

line 374: "(including variance of these accuracy)"

line 378-379: "The advancement in neural network architectures, and their power in potentially better adapting to
problem structure and input distribution in a data-driven manner, have brought new ways..."

- Another comment on the structure of the paper:  Currently, you explain the greedy discretization in the CNN-Baker section,
but since you also use it in GNN baker it would improve the clarity of the text if you described it in a separate paragraph.

- In the introduction, to motivate your approach you mention the existence of
real-world problems that involve geometric intersection graphs. Your experiments are
on synthetic randomly sampled instances. It would be nice to include experiments on real-world data.

- While we can see that the NN-Baker method performs well on large graphs, we are not given information about
how quickly it runs. Your framework seems to improve results across the board but it's not clear at what time cost this happens. Furthermore, showing results of NN-Baker on varying graph sizes like it was done in Figure 3
would help build a stronger case for your framework.

- In the paper, you claim that your framework extends to other problems like maximum acyclic subgraph.
 Having experimental results on at least one of them would help improve the paper's experimental section, both
in terms of novelty and in terms of providing evidence for your theoretical results.

- Minor detail regarding Theorem 3.2: The result from 3.1 gives you a universal approximation result for a function
that maps a vector to a real number, while the function you want in Step 2' of NN-Baker
is $g_\mathcal{N}: [0,1]^k \rightarrow [0,1]^k$. The result from 3.1 extends to vector-valued outputs but it should
be explicitly stated for the sake of clarity.

- An ablation study on NN-Baker would also help illustrate its benefits over the non-neural version.
  For instance, is it possible to compare with the non-neural Baker framework, or is it too slow because of the brute force enumeration?
  Could you replace the subset enumeration in step 2 with a maximum independent set heuristic? I understand that you
  may lose the approximation guarantee that way, but heuristics can perform very well so it is important
 to understand whether the neural networks are indeed beneficial in practice.

- In line 212, you claim that in practice you can start adding the points that were close to the grid back to the solution as long
as they don't violate your condition. Do you actually do this in your experiments? If you do, it would be good to have an ablation study that shows the contribution of this "post-processing" step in your results.

### Recap:

Overall, I find the contributions and the direction of the paper interesting.  Theoretical guarantees are particularly appealing
in the context of machine learning.
However, given that the paper's scope is already limited on a particular class of graphs and a specific subset of combinatorial problems, I believe that more thorough experimental demonstrations are necessary (I provided some examples) to make a strong case for this method.  Finally, some refinement is necessary in terms of writing in order to improve the clarity and quality of the paper.

For now, I will give the paper a 5 but I am willing to adjust my grade after the rebuttal.




**Time Spent Reviewing:**

12

---

> ### Author Response · Authors · 2021-08-10
> **Response to Reviewer JP57**
>
> Thank you very much for your time and all your comments. We will incorporate many comments into the revised version of this paper accordingly. Below we will respond to some of your major comments.
>
> First, we are glad that you appreciate the novelty of the overall Baker-NN framework which integrates algorithmic insights with ML frameworks. We believe that the high-level idea of decomposing an input problem into small subproblems algorithmically and then using ML approaches to tackle these subproblems, is general and applicable to many problems beyond what’s currently studied in our submission -- we will elaborate this more below in our response in bullet point (2). Our present paper demonstrates the utility of such an approach and makes a first step in this direction. In the future we will aim to solve other problems using such algorithmic decomposition + ML subproblems ideas.
>
> (1) Regarding “While we can see that the NN-Baker method performs well on large graphs, we are not given information about how quickly it runs. Your framework seems to improve results across the board but it's not clear at what time cost this happens. Furthermore, showing results of NN-Baker on varying graph sizes like it was done in Figure 3 would help build a stronger case for your framework. ”
>
> Response: Thank you for your comment. This is a valid point and we should include such timing information. Note that for NN-Baker architectures, there is both the training time and also testing time.
> The training time needed for each of the three types of 2D datasets of our NN-Baker architectures and their original non-Baker versions are as follows ('h’ stands for 'hours’):
>
> | Erdos | ErdosBaker | TGS | TGSBaker | LwD | LwDBaker | UNetBaker |
> |-------|------------|-----|----------|-----|----------|-----------|
> | 4h    | 7h         | 8h  | 8h       | 11h | 10h      | 7h        |
>
> However, once trained, we can compute a independent set of each test graph very efficiently: in particular, the running time is only around 20-80 seconds for each test graph. (The time needed for the non-Baker version of these neural networks, such as LwD, is only slightly faster than their Baker version.) Please see the detailed results reported in our response to Reviewer YGgd (bullet point (1)). In contrast, as we reported in our response to Reviewer YGgd, if we consider a Baker + Heuristic version, more precisely, what we call KaMIS-Baker, even with time 5 - 10 times longer than our NN-Bakers, KaMIS-Baker still achieves a worse accuracy.
> Furthermore, we would be happy to include a plot like Figure 3 to show the dependencies of the performance of our NN-Bakers w.r.t. graph size in the Supplement in our revised version. In general, we expect the performance depends more on the type of datasets (distribution of points) than on the number of points -- intuitively, this is because similar points distribution will give rise to similar distributions of bounded-size subproblems which an NN will learn to solve, regardless of the size of input graphs. From this point of view, there is no need to re-train NN-Baker for the same types of graph inputs. To test this, we performed the following: We generate two sets of graphs of around 14K-18K nodes from the 2D-dense distribution and the 2D-sparse distribution, respectively. Call these small-2D-dense and small-2D-sparse. The results reported in the first two rows of Table 1 of our paper are for graphs of around 40K-50K nodes. We simply use the same GNNs trained for handling the large graphs in our GNN-Baker architectures to test  these small graphs. The results are as follows: For small-2D-dense, the accuracy for ErdosBaker, TGSBaker and LwDBaker is 0.927, 0.947, and 0.957 respectively, slightly better but close to the accuracy of 0.923, 0.936 and 0.955 for large dense-graphs. For small-2D-sparse, the accuracy for ErdosBaker, TGSBaker and LwDBaker is 0.922, 0.939, and 0.934 respectively, also slightly better but close to the accuracy of 0.917, 0.931 and 0.926 for large dense-graphs.
>
> (2) Regarding the general scope of our proposed Baker-NN:
>
> Response: As we explained in the response to reviewer YGgd (bullet point (2)), Baker’s technique has been studied extensively within the algorithms community, and has been generalized to several other problems and input classes (Euclidean point sets, points from more general metric spaces, minor-free graphs, etc). Typically, at the heart of such an extension lies a domain-specific scheme for partitioning the input space into smaller subspaces. Our work initializes and constitutes a first step for this direction. We believe that this is a fruitful direction for generalizing our neural network-infused algorithmic approach to other domains, including graphs induced by points in more general metric spaces, or to tackle general graph problems by decomposing graphs to subgraphs with special structures (e.g, bounded treewidth, which has been widely used in developing efficient approximation or Fixed-Parameter-Tractable algorithms). Another promising future direction along this line is to further simulate the decomposition step with a well-designed neural network architecture (with theoretical properties).
>
> (3) Regarding your comment “An ablation study on NN-Baker would also help illustrate its benefits over the non-neural version. For instance, is it possible to compare with the non-neural Baker framework, or is it too slow because of the brute force enumeration? Could you replace the subset enumeration in step 2 with a maximum independent set heuristic?”
>
> Response: Thank you for your comment. Reviewer YGgd also has a similar comment (see bullet point (1) of our response to Reviewer YGgd). This is a very valid point, and we will include such comparisons. We have already carried out some experiments on Baker + KaMIS (where Step 2 is solved by KaMIS, a SOA heuristic approach for Max-independent set problem). Please see bullet point (1) in our response to Reviewer YGgd for details. Roughly speaking, with the similar running time, KaMIS-Baker performs much worse than NN-Baker. In order to achieve an accuracy close to our TGSBaker architecture, KaMIS-Baker needs more than 10 times running time.
>
> (4) Regarding your comment “In line 212, you claim that in practice you can start adding the points that were close to the grid back to the solution as long as they don't violate your condition. Do you actually do this in your experiments? If you do, it would be good to have an ablation study that shows the contribution of this "post-processing" step in your results.”
>
> Response: Yes, we indeed do this in all our experiments (consistently across all experiments, including KaMIS-Baker we newly experimented with). Thank you for the suggestion and we will include this ablation study. We have already collected some results: Roughly speaking, for our GNN-Baker architectures, our post-processing to add these additional valid points to the solution results in an improvement between 0.5 and 2.2 percentage points depending on the dataset and the model. For the UNetBaker model, we see a difference of 7.7 percentage points. It is interesting that more boundary points can be added back for UNetBaker (which is CNN based). We think part of the reason could be that on each tile, our CNN architecture may be less effective for pixels (points) close to the cell boundary (indeed, the likelihood map output by our UNet architecture seems to give lower likelihood around boundaries). In contrast, GNN is graph based and does not have this “boundary” effect.

---

> > ### Comment · Reviewer_JP57 · 2021-08-13
> > **Review update**
> >
> > My concerns regarding the experimental section of the paper have been partly addressed by the authors in their responses to me and the rest of the reviewers. The authors have also raised some valid points regarding the general research directions that this paper points to.
> >
> > This is a solid paper that presents an interesting take on combinatorial ML. Given the new information/experiments from the author responses (which should be appropriately added in the paper/supplement), I have updated my score.

---

> > > ### Author Response · Authors · 2021-08-13
> > > **Reply**
> > >
> > > Thank you for your time and comments. We will be sure to include all the new material / experiments in any revision of the paper.

---

### Official Review · Reviewer_YGgd · 2021-07-16

**Rating:** 6
**Confidence:** 3

**Summary:**

A machine learning framework for constructing heuristics for certain geometric graph optimization problems is proposed. Such graphs are derived from points on a 2d grid, for example. This geometric structure is typically useful for the design of approximation algorithms. This insight inspires the main question addressed in this paper: does the geometric structure also help improve ML approaches for graph optimization problems such as Maximum Independent Set (MIS), both theoretically and practically?

The authors answer affirmatively by leveraging an existing algorithmic paradigm from Baker (1994). The key idea is to decompose the space into a number of independent subproblems (linear in the graph size) of bounded sizes, then heuristically solving those small subproblems using an ML model. The ML model may be a convolutional network that see the subproblem is a pixel grid, or a graph neural network that acts directly on the small graph. The ML model may also be trained by supervision or without.

Theoretically, it is shown that if the ML model has universal approximation guarantees (e.g., 1-layer neural network), meaning that it can accurately approximate the function mapping from subproblem to subproblem optimum, then one obtains a bicriteria approximation guarantee on the global optimization solution, which is assembled appropriately from subproblem solutions.

Experimentally, it is shown that this decomposition approach can be combined with three different ML model classes, improving their original performance on a number of random problem families for MIS and vertex cover.


**Main Review:**

A machine learning framework for constructing heuristics for certain geometric graph optimization problems is proposed. Such graphs are derived from points on a 2d grid, for example. This geometric structure is typically useful for the design of approximation algorithms. This insight inspires the main question addressed in this paper: does the geometric structure also help improve ML approaches for graph optimization problems such as Maximum Independent Set (MIS), both theoretically and practically?

The authors answer affirmatively by leveraging an existing algorithmic paradigm from Baker (1994). The key idea is to decompose the space into a number of independent subproblems (linear in the graph size) of bounded sizes, then heuristically solving those small subproblems using an ML model. The ML model may be a convolutional network that see the subproblem is a pixel grid, or a graph neural network that acts directly on the small graph. The ML model may also be trained by supervision or without.

Theoretically, it is shown that if the ML model has universal approximation guarantees (e.g., 1-layer neural network), meaning that it can accurately approximate the function mapping from subproblem to subproblem optimum, then one obtains a bicriteria approximation guarantee on the global optimization solution, which is assembled appropriately from subproblem solutions.

Experimentally, it is shown that this decomposition approach can be combined with three different ML model classes, improving their original performance on a number of random problem families for MIS and vertex cover.


Overall, there is no shortage of creative algorithmic work in this paper. The experiments demonstrate a clear benefit to the proposed decomposition strategy compared to ML on the original full graph. However, the experiments do not go further to demonstrate that the NN-Baker approach can outperform the pure Baker approach without any NN. In particular, the NN replaces enumeration within each subproblem. Instead of enumeration, one could use integer programming through branch and bound (which is only enumeration in the very worst case, and empirically performs much better than that). Alternatively, one could use the (learnable) greedy heuristic for MIS of Gupta, Rishi, and Tim Roughgarden. "A PAC approach to application-specific algorithm selection." SIAM Journal on Computing 46.3 (2017): 992-1017. What I'm trying to say is that while Baker-NN can improve over NN, it is not clear that it can improve over Baker-ILP or Baker-(other types of ML). Because adapting the Baker approximation framework to new combinatorial problems (on geometric graphs) requires some serious algorithmic work, I suspect that the proposed framework has limited applicability beyond the problems studied herein.

Nonetheless, the paper is of high quality and so I lean towards accepting. I am eager to hear the authors' thoughts on my criticisms.

Other questions/comments:
- Baker paradigm: The way Algorithm Baker-MIS is presented at the moment, it is hard to tell how much of it is known from Baker's paper. Some discussion of exactly how you are extending/modifying/specializing the Baker paradigm would be good.

- Experimental results: you report the approximation ratio of a method's MIS to the optimum. This optimum respects the unit-ball constraint, whereas a learning method's MIS might not. How is that factored into the reported ratios? Am I missing anything?

Minor:
- line 6: "algorithm that is linear in graph size" --> "algorithm that runs in time linear in the graph size"
- line 13 "approximately solve"; this is only true if the NN being used is actually a universal approximator, which makes me think that the claim in this phrase is a bit too strong.
- line 84 "exponential in terms of the parameters", what parameters?
- Figure 1, "paradigmfor" --> "paradigm for"
- Figure 1, "$X_C/X_C^{'}", should that be a backward slack instead to denote set difference?
- line 159, shouldn't it be $\alpha > 1$, as in line 164?
- line 160, the cardinality of the set $Y$ should be used in the in the inequalities, as you do in line 166.
- lines 161-166: it would be good to articulate the definition of the bi-criteria approximation guarantee in words as well, "The algorithm is guaranteed to return an $\alpha$-approximate solution to the problem, while the unit-ball constraint by a factor of at most $\beta$."
- Algorithm Baker-MIS (page 5) is a bit difficult to parse. There is lots of notation and the figure is good but too simplified to allow for a reimplementation. Pseudocode or a clearer exposition in the appendix might be worthwhile.
- line 217: "Detials" --> "Details"

**Time Spent Reviewing:**

3

---

> ### Author Response · Authors · 2021-08-10
> **Response to Reviewer YGgd**
>
> Thank you very much for your time and all your comments. We will incorporate many comments into the revision accordingly. Below we will respond to the major comments.
>
> First, we are glad that you appreciate the novelty of the overall Baker-NN framework which integrates algorithmic insights with ML frameworks. We believe that the high-level idea of decomposing an input problem into small subproblems algorithmically and then using ML approaches to tackle these subproblems is general and applicable to many problems beyond what’s currently studied in our submission -- we elaborate on this more in bullet point (2).
>
> (1) Regarding your comment “However, the experiments do not go further to demonstrate that the NN-Baker approach can outperform the pure Baker approach without any NN.”
>
> Response:  That is a fair point and we will include such comparisons in the revision. Previously, we didn’t include such comparisons as we were comparing with SOA NN-based approaches and showing that the Baker-versions of them can improve them. Due to the time limit, we carry out the following experiments (but we will conduct also experiments on Baker + exact algorithm based on say branch and bound and include in the revision as well): In particular, we take a SOA heuristic approach for max-independent set (MIS) problem, called KaMIS (Ref [39] in the paper). We use KaMIS to solve each subproblem in the Baker framework (instead of using NN), and call the resulting method KaMISBaker. For the KaMIS approach, one can adjust the time budget and longer time can lead to better results. Below we will use 3 versions of KaMISBaker, denoted by KaMISBaker-1, KaMISBaker-2, and KaMISBaker-3, with three different running times. We use the time and accuracy for UNetBaker, TGSBaker and LwDBaker for comparison. Note that NN-Baker uses GPUs, while KaMISBaker uses only CPUs. To make it more fair for Baker-KaMIS, we deploy 16 CPUs (while a single RTX-A6000 GPU is used for NN-Baker results). For each data set, the running time is the average running time for 100 test graphs.
>
> | Data        | UNetBaker | TGSBaker | LwDBaker | KaMISBaker-1 | KaMISBaker-2 | KaMISBaker-3 |
> |-------------|-----------|----------|----------|--------------|--------------|--------------|
> | 2D-dense    | 18s       | 15s      | 44s      | 78s          | 138s         | 415s         |
> | 2D-sparse   | 60s       | 33s      | 81s      | 75s          | 127s         | 402s         |
> | 2D-gaussian | 58s       | 28s      | 62s      | 82s          | 135s         | 420         |
>
> The performance of KaMISBakers vs NN-Bakers are:
>
> | Data        | UNetBaker | TGSBaker | LwDBaker | KaMISBaker-1 | KaMISBaker-2 | KaMISBaker-3 |
> |-------------|-----------|----------|----------|--------------|--------------|--------------|
> | 2D-dense    | 0.915| 0.936| 0.955| 0.753| 0.816| 0.914|
> | 2D-sparse   | 0.919| 0.931| 0.926| 0.672| 0.784  | 0.903        |
> | 2D-gaussian | 0.917     | 0.927    | 0.925    | 0.674        | 0.831        | 0.912        |
>
> To achieve an accuracy that is close to our NN-Baker architectures, KaMISBaker has significantly longer running time (more than 10 times slower than our TGSBaker, or 5-10 times slower than our LwDBaker). We do note that the performance of our NN-Bakers comes at a price. We must train the neural networks between 7 -14 hours depending on the model and data. Once trained, the NN-Baker is efficient on new graphs.  Some results of applying a NN-Baker trained on one type of graphs but tested on different types of graphs are reported in the Supplement.
>
> (2) Regarding your comment “Because adapting the Baker approximation framework to new combinatorial problems (on geometric graphs) requires some serious algorithmic work, I suspect that the proposed framework has limited applicability beyond the problems studied herein.”
>
> Response: The Baker’s technique was originally developed to solve several optimization problems in planar graphs. By Baker’s technique, we refer to the key underlying idea of decomposing an input problem into small problems whose solutions can be stitched back into a solution for the global problem with approximation guarantees. This idea has broad applicability. Indeed, since the original paper (Ref [28]), this idea has been successfully extended to other problems (e.g, TSP problem in [Klein, SIAM J. Computing 2008]) and other settings, e.g., see [Arora, Mathematical Programming 2003] for a survey on extensions to geometric settings. The Baker’s technique has also been adapted to metric spaces of bounded intrinsic dimension (e.g., [Bartal et al., SIAM J. Computing 2016]), and to minor-free graphs (e.g., [Tazari, Sym. Mathematical Foundations of Computer Science 2010]). One of the main contributions of our paper is to show that this idea (originated by Baker) can be naturally integrated with neural networks so as to solve hard problems with theoretical guarantees. We believe that this is a fruitful direction for generalizing our neural network-infused algorithmic approach to other domains. Our paper serves as a first step in this direction and we aim to explore many other problems (e.g, TSP problem, or in more general metric space setting) in the future. We can also explore how to decompose input graphs into subgraphs with simpler structures (e.g, bounded treewidth, which has been widely used in developing efficient approximation or Fixed-Parameter-Tractable algorithms). Another interesting direction we hope to explore is to further simulate the decomposition step with a well-designed neural network architecture (with theoretical properties). We will add a discussion on this in our revision.
>
> (3) Regarding “Baker paradigm: The way Algorithm Baker-MIS is presented at the moment, it is hard to tell how much of it is known from Baker's paper. Some discussion of exactly how you are extending/modifying/specializing the Baker paradigm would be good.”
>
> Response: We will include such a discussion in the revision of our paper. Roughly speaking, the original Baker technique was applied to solve problems in the planar graphs.Typically, for different problems in different settings, at the heart of such an extension lies a domain-specific scheme for partitioning the input space into smaller subspaces, and prove theoretical guarantee after stitching. The most relevant work to our Baker-dMIS ([HW]: Hochbaum and Wolfgang, "Approximation schemes for covering and packing problems in image processing and VLSI." Journal of the ACM (JACM) 1985), where the authors obtain approximation algorithms for maximum independent set and minimum vertex cover on unid disks graphs. Compared to their algorithms, our results differ in the following ways:
> In order to make each subproblem amenable to a neural network solver (especially with bounded size), we discretize each cell into a set of pixels. The overall NN-Baker framework hence can handle input of different sizes / scales.
> The results in [HW] are obtained by carefully choosing the best shift for the underlying grid. In contrast, we pick the shift randomly.
> We will include a more detailed discussion of the related work in the final version of our paper.
>
> (4) Regarding “Experimental results: you report the approximation ratio of a method's MIS to the optimum. This optimum respects the unit-ball constraint, whereas a learning method's MIS might not. How is that factored into the reported ratios? Am I missing anything? ”
>
> Response:  Thank you for the comment, and we will clarify this in the revision. For our returned solutions from each version of the NN-Baker architectures, the unit-ball constraint is strictly enforced. In particular, note that in both CNN-Baker and GNN-Baker, the output is a likelihood that a pixel is in a MIS. When post-processing to produce a solution, we in fact use an arbitrary input point in each non-empty pixel as its representative. Then during the greed,y iterative approach as described in lines 309--315 of our submission, we use these representatives to check whether the pairwise distance is at most 2 or not. Hence the output set of points from $X_C$ are guaranteed to be at least distance 2 apart from each other. The description in lines 309-315 of our submission uses the center-point of each pixel to first general $Z’$, and then relaxes each pixel in $Z’$ to a point in $X_C$. Using this approach, the output set may violate distance threshold while still satisfying the bi-criteria approximation. In our experiments, we used the aforementioned approach to compute our output and thus we always output a valid independent set. We should and will clarify this in the submission.To see the difference between these two approaches, we have also implemented the exact approach as described in lines 309--315, and the output differs very little (around 0.001).
>
> (5) Regarding “Alternatively, one could use the (learnable) greedy heuristic for MIS of Gupta, Rishi, and Tim Roughgarden. "A PAC approach to application-specific algorithm selection." SIAM Journal on Computing 46.3 (2017): 992-1017. ”
>
> Response: Thank you for the reference. The main (meta-)algorithm in this paper is an Empirical Risk Minimization (EMR) algorithm, which picks an approximately optimal heuristic from a given set of heuristics, for a fixed but unknown input distribution. This is used to tune the parameters of two different heuristics for Maximum Independent Set (based on greedy and local search). Unfortunately, the best possible implementation of the resulting algorithm has running time at least Omega(n^4) for greedy and Omega(n^8) for local search, where n is the number of vertices. Therefore, these methods are mostly of theoretical interest, and not applicable to our setting. We also emphasize that, even if such a method was applicable, the guarantee of this paper is only optimality within the given collection of heuristics. Thus there is no theoretical reason to expect that the computed independent sets would be nearly-optimal.

---

> > ### Comment · Reviewer_YGgd · 2021-09-01
> > **Great**
> >
> > I appreciate the thorough response! I stand by my score and hope the paper will be accepted.

---

> > > ### Author Response · Authors · 2021-09-01
> > > **Thanks**
> > >
> > > Thanks for your time and comments!

---

### Decision · Program_Chairs · 2021-09-27

**Decision:**

Accept (Poster)

**Comment:**

All reviewers found the papier interesting and appreciated the articulation with theoretical guarantees.
The reviewers have asked a number of questions and made a number of comments: the authors are strongly encourage to take them into account and to use the elements that they provided in their responses to the reviewers to prepare the final version of the paper.